# Distribution of Human Papillomavirus Genotypes among the Women of South Andaman Island, India

**DOI:** 10.3390/diagnostics13172765

**Published:** 2023-08-25

**Authors:** Rehnuma Parvez, Paluru Vijayachari, Mrinmoy Kumar Saha, Lipika Biswas, Jawahar Ramasamy, Alwin Vins, Nisha Beniwal, S. Vasanthi, Sasikala Ramadoss, Harpreet Kaur, Muruganandam Nagarajan

**Affiliations:** 1ICMR—Regional Medical Research Centre, Port Blair 744103, India; directorrmrc@gmail.com (P.V.); lipikabiswas60@gmail.com (L.B.); vinsav2@gmail.com (A.V.); beniwalnisha7@gmail.com (N.B.); 2G. B. Pant Hospital, ANIIMS, Port Blair 744104, India; mksaha@rediffmail.com; 3Aarupadai Veedu Medical College and Hospital, Vinayaka Mission’s Research Foundation (DU), Pondicherry 607402, India; jawahar84@gmail.com; 4Vinayaka Mission Research Foundation, Salem 636308, India; vasamthifom@gmail.com (S.V.); sasi2508@yahoo.com (S.R.); 5Indian Council of Medical Research—Headquarters, New Delhi 110029, India; kaurh.hq@icmr.gov.in

**Keywords:** HPV, genotypes, lineages, India, women, HR-HPV Andaman and Nicobar Islands

## Abstract

Background: Human Papillomavirus (HPV) causes various types of cancer in both men and women. Woman with HPV infection has a risk of developing invasive cervical cancer. Globally, HPV 16 and 18 were predominant. This study aims to find the distribution of various HPV types in South Andaman. Methods: A cross-sectional study was conducted among women in South Andaman, where cervical scrapes were collected after collecting written informed consent. Detection of HPV genotypes was carried out by using a PCR assay. Further, sequencing analysis was performed using MEGA11 to identify various genotypes in this territory. Result: Of these 1000 samples, 32 were positive for HR-HPV 16, and four were positive for HR-HPV 18. Fifteen HPV genotypes were detected using molecular evolutionary analysis. Six cases were identified with multiple genotypes. The most prevalent genotype is HPV 16 which belongs to Lineage-A and sub-lineage A2. HPV 18 identified in South Andaman belonged to the lineage A1 to A5. Discussion: Various HPV types were identified among women in South Andaman. Global burden of cervical cancer associated with various HPV sub-lineages. HPV-16 A1 sub-lineage was globally widespread, whereas sub-lineages A1, A2 and D1 prevailed in South Andaman. Conclusions: HR-HPV identified in this study enlightens the importance of HPV vaccination among women in remote places. These findings will help to strengthen public health awareness programs and prevention strategies for women in remote areas.

## 1. Background

HPV is responsible for most human reproductive tract viral infections. Most HPV infections are asymptomatic and self-limiting; chronic infections can progress into warts in precancerous, cervical, anogenital or oropharyngeal regions in men and women. Cervical cancer was the most frequent HPV associated disease. Though the majority of HPV pre-cancerous lesions have a tendency to disappear on their own, there remains a risk for every woman with HPV infection to become persistent and pre-cancerous leading to invasive cervical cancer [1].

Cervical cancer is the leading cause of death in women. According to Global Cancer Observatory (GLOBOCAN) 2020, the incidence rate of cervical cancer was 15.6%, and the mortality rate was 8.8% worldwide. The age-specific standardized rate of cervical cancer was 13.3%. In Asia, the incidence rate of cervical cancer was 58.2 per cent. The five-year prevalence of cervical cancer was 59.5% in Asia. In India, the incidence and mortality rates were 16.2% and 9.5%, respectively. The proportion of cervical cancer in India was 7.9 per 100,000 [2]. In India, 22% of women have undergone cervical screening examinations based on aNational Family Health Survey (NFHS) report [3]. According to the World Health Organization (WHO), 99% of cervical cancer cases were associated with high-risk human papillomavirus (HR-HPV) [1]. Among the Indian population, the prevalence of cervical cancer was higher among sex workers in an urban slum in Mumbai and HIV-positive women. HPV 16 and 18 were observed in 56% of cases in the West Indian region [4,5]. HPV belongs to the family Papillomaviridae family, and it is a small, non-enveloped circular double-stranded DNA virus. The DNA molecule is 8000 base pairs in size, and the genome has six early (E) E1-E2, E4-E7 regions and two late (L) L1 and L2 regions [6].

Papillomavirus is classified into alpha-papillomavirus, beta-papillomavirus, gamma-papillomavirus, delta-papillomavirus, and zeta- papillomavirus and theta- papillomavirus. The alpha and beta papillomaviruses were commonly detected in human samples. Alpha-papillomavirus causes mucosal and cutaneous lesions. Based on Molecular data, further alpha-papillomavirus is classified as low-risk and high-risk. HPV-32, HPV-10, HPV-61, HPV-76, HPV-54, and HPV-71 are causing low-risk benign tumours(mucosal lesions and cutaneous lesions). However, HPV-2, HPV-26, HPV-34,HPV-16,HPV-18,and HPV-53 are responsible for high-risk mucosal lesions. Beta-papillomavirus cause malignant cutaneous lesions. Among beta-papilloma viruses, HPV 5 and 9 were most commonly associated with benign lesions in immunosuppressed patients. HPV 4, 48, 50, and 60 were responsible for cutaneous lesions. Other papillomaviruses, such as Delta-papilloma virus, zeta papillomavirus, eta-papillomavirus, and theta-papillomavirus, were responsible for lesions in cattle, horses, and birds [7,8]. Over 200 different HPV genotypes have been characterised, conforming to the >10% diversity within the L1 gene sequence [9].

HR-HPV types are 16, 18, 31, 33, 35, 39, 45, 51, 52, 56, 58, 59, 68, 73, 53, 30, 66 and 82. Low-risk HPV (LR-HPV) types are 6, 11, 40, 42, 43, 44, 54, 61, 70, 72, 81, and CP6108. This epidemiological classification and the classification based on phylogenetic grouping were pretty close to each other. HPV 16 and HPV 18 are two particularly potent and obviously oncogenic [7,8]. Cervical cancer was attributable to one of 13 HPV types (HPV 16, 18, 33, 31, 45, 56, 35, 52, 56, 58, 59, and 68) as a cancer-associated agent, whereas 8 HPV types (HPV 26, 53, 66, 67, 68, 70, 73, and 82) that were phylogenetically related to 12 WHO-defined HR-HPV have consistently been identified as single HPV infections and categorised as potentially carcinogenic possible (p) HR-HPV [10].

Andaman & Nicobar Islands are situated in the southern regions of the Bay of Bengal in the Indian Ocean, closer to Indonesia and Thailand. According to the Census of India (2011), the territory’s population was 380,581, and the female population was 177,710 (46.7%). The literacy rate of the Andaman and Nicobar Islands is 77.3% [11].

Our previous study in Andaman and Nicobar Islands reported the HR-HPV types (HPV 16 and 18). This was the first of its kind study to find HPV types in these islands [12]. HPV variants were not studied in detail among the population in this region so far. It is necessary that the public health system should be aware of the circulating variants of local strain patterns of HPV to frame recommendations for developing appropriate broad-spectrumvaccines aiming at HR-HPV variants.To our knowledge, this study was the first cross-sectional survey conducted among a large population across Andaman and Nicobar Islands. Further, the current study aims to know the variants of HPV among married women in the Andaman and Nicobar Islands.

## 2. Methodology

### 2.1. Study Population

A community-based cross-sectional study was conducted among married women of reproductive age (18–59 years) residing in the South Andaman District of the Andaman and Nicobar Islands, India.

### 2.2. Exclusion Criteria

Patients were excluded if there was evidence of pregnancy, severe gynaecological bleeding, hysterectomy or previous history of the disease, including cancer, warts and other cutaneous manifestations.

### 2.3. Ethical Approval

This study has been approved by the Institutional Human Ethics Committee (IHEC) of the Indian Council of Medical Research—Regional Medical Research Centre (ICMR-RMRC), Port Blair [IEC No: 03/RMRC/29/06/2017].

### 2.4. Sampling and Sample Size

The target population was chosen via cluster sampling, and the sampling units were villages or municipal wards. After stratifying the sampling units into rural/urban strata, the required sample size was determined by random selection of the required sample size’s units. Based on the Andaman population ratio, the study participants were drawn from rural villages and urban wards in a ratio of 2.5:1, yielding a sample of 700 from the rural and 300 from the urban.

### 2.5. Awareness Programmes

Initially, awareness programmes were conducted in each selected village/ward at the Anganwadi centres/community hall. The health care team (clinician along with trained nurses) were detailed about the health issues like cervical cancer, its symptoms, and genital hygiene. In addition, the need for the study was also explained and requested for written informed consent before enrollment.

### 2.6. Sample Collection and Storage

The enrolled women were called to the field clinics, which were held in the sub-centre, Primary Health Centre (PHC), Community Health Centre (CHC) and District Hospitals catering for the population of the particular village/wards. The cervical scrapes were collected using the standard procedure from the ectocervix or surface of the cervical portion using a cytobrush.Specimens were collected in a tube containing phosphate-buffered saline (pH 8.6) and transported to the laboratory in ICMR-RMRC, Port Blair, by maintaining a cold chain.

### 2.7. Sample Processing

Once the samples arrived at the laboratory, the specimen tubes were vortexed, cytobrushes were discarded, and tubes were centrifuged to pellet the cells, which were suspended in 1 mL of phosphate-buffered saline. Aliquots of each fresh specimen were made and stored for a short duration at −20 °C until further processing.The analysis of HPV DNA was performed in the molecular biology laboratory of ICMR-RMRC, Port Blair.

### 2.8. DNA Extraction

The total DNA was extracted with the QIAamp DNA Minikit (Qiagen, Hilden, Germany) according to the manufacturer’s instructions. The DNA was eluted in 45 μL of elution buffer.

### 2.9. PCR Assays

The isolated DNA was amplified with ß-globin (internal control) to ensure the purity of the DNA extractions, as described previously [13].

To confirm the HPV infection, the DNA of all the samples were subjected to PCR amplification targeting the L1 consensus gene by a standard procedure reported previously. The results were recorded as positive if amplicon size specific to the 450 bp DNA band was observed in agarose gel electrophoresis.

### 2.10. Detection of HPV 16 & 18

PCR for the detection of type-specific HR-HPV 16 & 18 in the predominant genotypes was performed [13,14]. In addition, the E6 and E7 genes of HPV 16 and the E6 gene of HPV 18 were also amplified to identify the lineages and sub-lineages of HR-HPV 16 & 18 in the South Andaman Islands [14].

### 2.11. PCR Sequencing

The DNA sequence analysis was carried out to confirm the HPV types distributed in South Andaman. L1 gene PCR amplicons of all the samples negative for HPV 16 and 18.as well as 7 samples of HPV 16 confirmed, were subjected to DNA sequence analysis. In addition, the E6 gene and E7 gene PCR amplicons of HPV 16 and the E6 gene of HPV 18 were also subjected to DNA sequencing.DNA sequencing was carried out by the Sanger sequencing method with corresponding primer sets [15].

### 2.12. Phylogenetic Analysis

The DNA sequences were assembled using the MEGA11 software tool and were analysed together with worldwide diverse HPV sequences using ClustalW multiple alignments and pairwise alignment for phylogenetic analysis and were subsequently analysed using Kimura’s two-parameter model as a method of substitution and neighbour-joining to reconstruct the phylogenetic tree. The statistical significance of the relationships obtained was estimated by bootstrap resampling analysis (1000 repetitions). A similar analysis was performed for the E6 and E7 genes of HPV 16 and E6 gene HPV 18.

The phylogenetic trees depicting the evolutionary relationship between taxonomic groups were generated for L1 genes of HPVs, E6 and E7 genes HPV 16, and E6 gene of HPV 18 sequences using molecular evolutionary genomic analyser software MEGA 11 [16]. Genetic distances were calculated by using the Kimura 2 parameter (K2P) model at the nucleotide level, and phylogenetic trees were constructed by using the neighbour-joining method. The reliability of the phylogenetic trees was tested using the bootstrap test with 1000 bootstrap replications.

## 3. Results

All of the cervical samples tested positive for the β-globin gene, indicating that there were adequate cells in the samples. Out of 1000 samples screened, 50 specimens tested positive for HPV L1 gene amplification. Subsequently, type-specific PCR for HR-HPV 16 and 18 identified 32 patients positive for HR-HPV 16 and four patients positive for HR-HPV 18. DNA sequencing for the L1 region was successful for 24 samples. The molecular evolutionary genetic analysis of sequences from South Andaman and worldwide was performed, and the pairwise genetic distances between the closely related HPV types from worldwide are specified in Table 1, given below.

The HPVs found in South Andaman had genetic relatedness with various genotypes worldwide, according to the phylogenetic analysis of the HPV-L1 partial gene. Out of the 24 sequences of the L1 region, 29.2% (*n* = 7) were clustered with HPV 16, 8.3% (*n* = 2) with HPV 6, 4.2% (*n* = 1) with HPV 52, 4.2% (*n* = 1) with HPV 33, 8.3% (*n* = 2) with HPV 58, 4.2% (*n* = 1) with HPV 73, 8.3% (*n* = 2) with HPV 66, 4.2% (*n* = 1) with HPV 53, 4.2% (*n* = 1) with HPV 71, 8.3% (*n* = 2) with HPV 84, 4.2% (*n* = 1) with HPV 87, 4.2% (*n* = 1) with HPV 81, 4.2% (*n* = 1) with HPV 30, and 4.2% (*n* = 1) with HPV 61. A MEGA version 11 software tool-based Neighbour-joining phylogenetic analysis for the HPV-L1 partial gene exhibited the distribution of different HPV types in South Andaman Island (Figure 1).

Of these 7 HPV-16 cases, 5 (AN MT-19, AN MK-72, AN AH-11(i), AN AL-1, AN MH-36 were from rural areas and 2 (AN AH-210, AN PV-40(i)) were from urban areas. Among the other HR-HPV types, HPV-58 (AN WN-08, AN WN-34) and HPV-53 (AN TM-08) were observed in rural areas. However, HPV-33 (AN PV-03), HPV-52 (AN PV-06), HPV-73(AN AH-09) and 66 (AN AH-53, AN AH-88) were reported in urban areas of South Andaman district. Moreover, HPV-6, 61, 81, 87, and 84 belonged to LR-HPV and were reported in the rural areas of South Andaman (Table 1).

The distribution of the multiple genotypes of HPV in South Andaman could be determined by a combined analysis that uses a specific PCR and DNA sequence analysis. Table 2. lists the genotype, risk group, frequency, and percentage of HPV detected in South Andaman Island. The molecular evolutionary genetic analysis could identify the distribution of HPV types 16, 52, 58, 66, 33, 18, 73, 53, 30, 6, 61, 71, 81, 84 and 87. There were high-risk, as well as LR-HPV types prevalent among the women in South Andaman Island.

Among the 50 samples identified by the HPV L1 gene, six cases had HPV co-infections. One case was found to be co-infected with 3 HPVs 16, 18 & 71. Five cases were identified with co-infection with two HPVs, i.e., (16 & 18, 16 & 61, 16 & 84, 18 & 30 and 66 & 87). All the co-infected cases were found to have at least one HR-HPV type association. However, HPV type could not be identified in 2 samples due to exhaustion of specimens for repeated experiments.

### 3.1. Phylogenetic Analysis of E6 Gene

The majority of the HPV types distributed in South Andaman were found to be HR-HPV 16, followed by HR-HPV 18. The distribution of various lineages and sub-lineages in South Andaman Island was revealed by phylogenetic analysis of the HPV 16 partial E6 gene (Figure 2A).

The phylogenetic analysis revealed that thirteen of the fourteen sequences were found to be associated with lineage A, and the remaining one was associated with lineage D. The majority (11) of HPV 16 sequences were grouped with AF536179 which belongs to the sub-lineage A2 (K2P = 0.002%). The pairwise genetic distance between the two isolates from South Andaman (AN GP-20 and AN PV-59) and the European isolate (K02718) was found that the South Andaman isolates belong to the A1 lineage (0.000%). One isolate (ANMT-19) identified from South Andaman was grouped with the HQ644257, which belongs to the D1 lineage (K2P = 0.00%).

Analysis to identify the HPV 18 sub-lineages revealed that the partial E6 gene of HPV 18 had close genetic relatedness with reference sequences of HPV 18 Lineages A. The E6 gene of HPV 18 identified from South Andaman was associated with the lineage A1 to A5 (K2P = 0.00%) (Figure 2B). However, there were no genetic differences within the lineages of the E6 partial gene region sequenced to identify the sub-lineage.

### 3.2. Phylogenetic Analysis of E7 Gene

The phylogenetic analysis of the E7 gene of HPV 16 from the South Andaman district showed maximum genetic relatedness with lineage A (Figure 3). Hence the predominant lineage circulating in South Andaman was identified as lineage A. However, the analysis based on genetic distances between the sequences did not show apparent sub-lineage differentiation in the E7 gene, as seen in the E6 gene of HPV 16.

## 4. Discussion

The current study provided the diversity of HPV among women in South Andaman Island. It is essential to comprehend the spectrum of HPV genotypes because data on the distribution of HPV genotypes are relevant to vaccine development. HPV 16 and HPV 18 cause more than 70 percent of cervical cancer cases, with the remaining cervical cancers caused by other HR-HPV genotypes [17].

Globally, the most prevalent genotypes were HPV-16 and 18. Other HR-HPV genotypes were HPV-59, 73, 45, 31, 33, 52, 35, 39, and 68 [18]. The prevalence of HPV type 16 and 18 in Asia is 55.1% and 13.8%. Other types, such as 31, 33, 35, 39, 52, 58, and 59, were also reported in Asia. HPV prevalence was associated with invasive cervical cancer [19]. In India, the prevalence of cervical cancer associated with HPV 16 and HPV 18 is 69.7% and 13.5%. Prevalence of other HR-HPV types, such as HPV 45, 33, 35, 58, 31, 59, and 52, is 5.1%, 4%, 2.5%, 2.4%, 2.3%, 1.4%, and 1.4% [20].

HPV 16 was the most prevalent strain in Central India. The other prevalent high-risk HPV types were types 18, 31, 35, 45, 56, and 59. Oral and vulva/vagina malignancies were exclusively related to HPV 16 [21]. Another study from Madhya Pradesh detected HPV 16 as a highly prevalent type found among ICC cases, followed by HPV 18, 45, 66, 35, and 56. Less prevalent types were HPV 31, 51, 58, 59, 67, 82, and JEB2 [22]. In this current study, the phylogenetic analysis revealed the presence of 15 HPV genotypes in the South Andaman District, including High-Risk types 16, 18, 33, 52, 53, 58, 66 and 73 and Low-Risk types 6, 30, 61, 71, 81, 84 and 87.

HPV 16 genetic variation may have a significant impact on cervical cancer risk. However, the global burden of cervical cancer associated with various sub-lineages is predominantly driven by past HPV 16 sub-lineage distribution. HPV-16 A1 sub-lineage was globally widespread. However, sub-lineages A3 and A4 were common in Asia. Sub-lineages A3, A4 and lineage D were common in regions like East Asia and North America.

Sub-lineage A4 was associated with more severe disease status than A1-3 sub-lineages in Chinese females and a higher risk of cancer. These lineages were highly cancer-risk associated [23]. In addition, lineage A of HPV 16 was found to be the prevailing strain in Spain. Lineage D of HPV 16 was linked to a higher risk of CIN3+ and otherhigh-grade lesions [24]. A study conducted in Eastern India revealed the existence of A1, D1 and D2 lineages. Of these lineages, A1 sub-lineage was predominant among women with cervical carcinoma [25]. The previous studies in India revealed the HPV 16 A1 (European) sublineage was predominant among cervical carcinoma patients compared to D1 (North American) and D2 (Asian-American-1) [15,25]. The current study found sub-lineages A1, A2, and D1 to be prevailing in South Andaman. The majority of the isolates from the current study belonged to the A2 sub-lineage.

A study revealed that the A1 sub-lineage of HPV 18 was predominant in Central Asia, Northern America and Eastern Asia. Nevertheless, A2 sub-lineages of HPV 18 were predominant in Europe, North America, Northern Africa and South/Central Asia. In addition, B1 and B2 sub-lineages were predominant only in Sub-Saharan Africa. Further, C-lineages were also observed in the African region [26]. HPV 18 sub-lineage distribution in China belonged to A1 to A7 [27,28]. Another study from Iran found that the prevalence of sub-lineage A4 was high compared to other sub-lineages [29]. In Spain, the study revealed Lineage B of HPV 18 was related to the burden for CIN3+ compared with lineage A [24]. In the current study, the South Andaman isolates belonged to lineage A of HPV 18.

The identification of the genetic underpinnings responsible for the distinct carcinogenic properties exhibited by certain lineages of HPV 16 and HPV 18 has the potential to shed light on the intricate interactions between the viral agents and the human host. Such insights hold promise for enhancing our ability to effectively manage HPV infections and mitigate the incidence of cervical cancer.

Diversity in the distribution of HPV types gives rise to a challenge to vaccine strategies. Molecular surveillance of HPV is needed for the detection of new strains or types emerging among symptomatic and asymptomatic populations in these remote islands. This will help policymakers to implement preventive measures against HPV-associated cervical cancer. A study on the specificity of HPV variants will be helpful in developing broad-spectrum vaccines aiming at HR-HPV variants.

## 5. Conclusions

This is the first-ever community-based cross-sectional study conducted in the Andaman and Nicobar Islands which interestingly revealed the prevalence of a wide range of the genotype distribution of HPV among women in this small island. HPV 16 was the most predominant high-risk type found in the Andaman Islands. Various high and low-risk types were also revealed in this study. Phylogenetic analysis of the E6 gene found lineages A and D of HPV 16 in the Andaman Islands. Moreover, lineage A of HPV 18 was also identified through the phylogenetic analysis.Sequencing analysis of the HPV 16 E6 gene revealed that A2 sub-lineages of HPV 16 were predominantly reported as compared to other lineages. The findings in the current study provide sufficient data to highlight the importance of screening for cervical cancer and promote vaccination and vaccine awareness in women living in remote geographical locations. These findings also emphasise and help to initiate stronger public health awareness programs and prevention strategies for the women of the Andaman and Nicobar Islands.

## Figures and Tables

**Figure 1 diagnostics-13-02765-f001:**
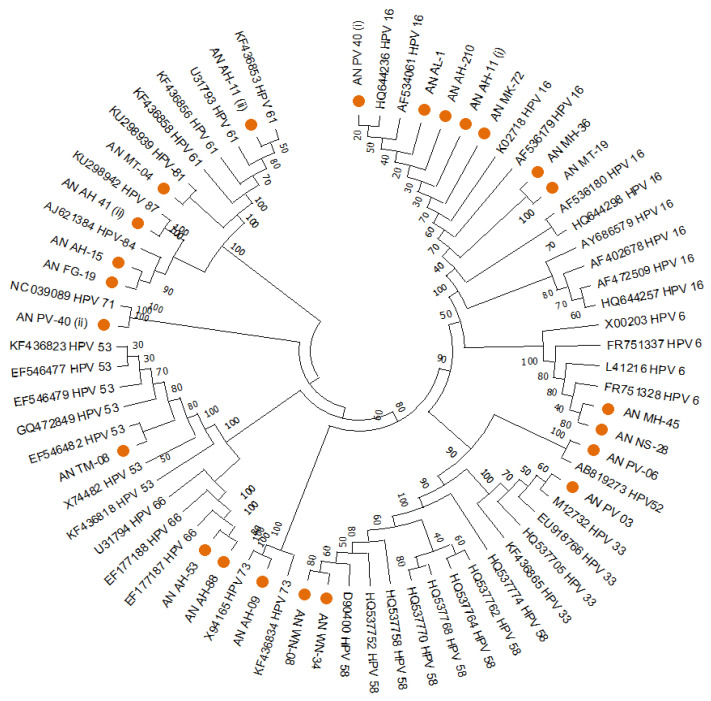
Phylogenetic analysis for the HPV-L1 partial gene displayed the distribution of different HPV types in South Andaman Island. All the sequences were analysed in MEGA11. The evolutionary history was inferred using the Neighbour-Joining method. (i) and (ii) indicates two types of HPV identified in the same sample.

**Figure 2 diagnostics-13-02765-f002:**
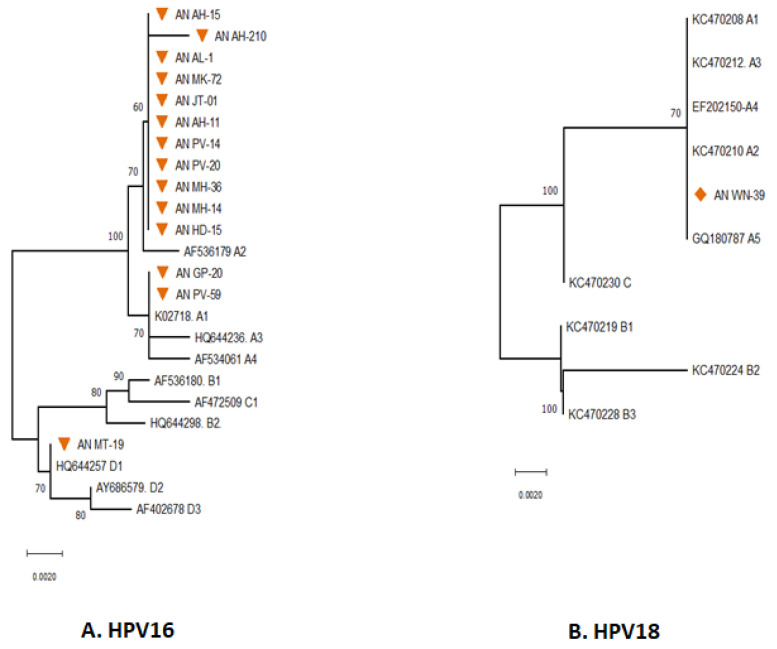
Phylogenetic analysis for the E6 partial gene displayed the distribution of different Lineages of HPV 16 (**A**) & 18 (**B**) in South Andaman Island. All the sequences were analysed in MEGA11, and the evolutionary history was inferred using the Neighbor-Joining method. An orange-coloured triangle indicates the HPV-16 isolates identified in South Andaman. Orange-coloured rhombus indicates the HPV-18 isolates identified in South Andaman.

**Figure 3 diagnostics-13-02765-f003:**
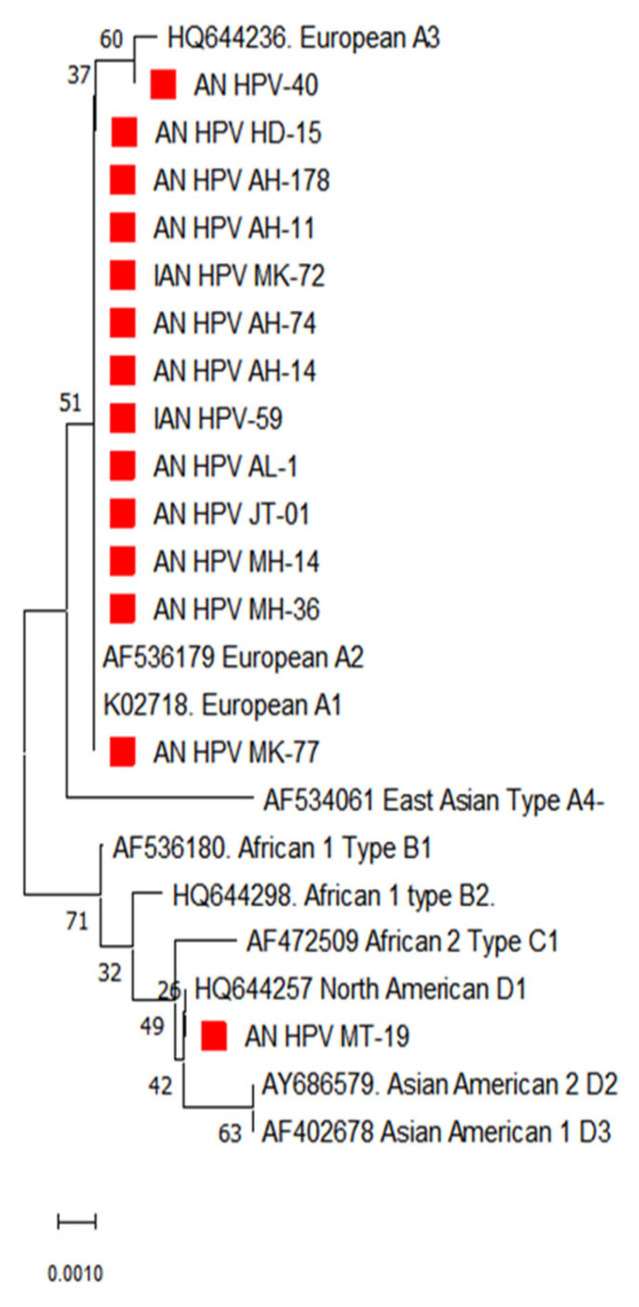
Phylogenetic analysis for the E7 partial gene displayed the distribution of different lineages of HPV-16 in South Andaman Island. The red square indicates the HPV-16 isolates identified in South Andaman.

**Table 1 diagnostics-13-02765-t001:** The pairwise genetic distances between the HPV types in South Andaman Island and closely related HPV types worldwide.

*S.No*	*Lab. ID*	*Urban/Rural*	*HPV Genotype*	*Genetically Close Related Worldwide Reference Sequences of HPV Types*		*K2P Distance Value*
				*Accession Number*	*Country Origin*	
** *HR-HPV Types* **						
1	AN-MT-19	Rural	HPV 16	AF472509	NY, USA	0.035
2	AN-MK-72	Rural	HPV 16	AF534061	NY, USA	0.002
3	AN-AH-11(i)	Rural	HPV 16	AF534061	NY, USA	0.000
4	AN-AL1	Rural	HPV 16	AF534061	NY, USA	0.000
5	AN-PV-40(i)	Urban	HPV 16	AF534061	NY, USA	0.000
6	AN-AH-210	Urban	HPV 16	AF534061	NY, USA	0.000
7	AN-MH-36	Rural	HPV 16	K02718	Britain	0.022
8	AN-PV-03	Urban	HPV 33	M12732	NY, USA	0.007
9	AN-PV-06	Urban	HPV 52	AB819273	Japan	0.000
10	AN-WN-08	Rural	HPV 58	D90400	Osaka, Japan	0.004
11	AN-WN-34	Rural	HPV 58	D90400	Osaka, Japan	0.004
12	AN-AH-53	Urban	HPV 66	EF177188	NY, USA	0.017
13	AN-AH-88	Urban	HPV 66	EF177188	NY, USA	0.022
14	AN-AH-09	Urban	HPV 73	X94165	Germany	0.000
15	AN-TM-08	Rural	HPV 53	EF546482	NY, USA	0.000
* **LR-HPV types** *						
16	AN-MH-45	Rural	HPV 6	FR751328	Slovania	0.009
17	AN-NS-28	Rural	HPV 6	FR751328	Slovania	0.002
18	AN-AH-32	Urban	HPV 30	KF436837	NY, USA	0.028
19	AN-AH-11(ii)	Rural	HPV 61	U31793	Los Alamos, NM, USA	0.00
20	AN-PV-40(ii)	Urban	HPV 71	NC039089	Tokyo, Japan	0.002
21	AN-MT-04	Rural	HPV 81	KU298939	Brazil	0.039
22	AN-FG-19	Rural	HPV 84	AJ621384	Cyprus	0.053
23	AN-AH-15	Rural	HPV 84	AJ621384	Cyprus	0.00
24	AN-AH-41(ii)	Rural	HPV 87	KU298942	Brazil	0.00

(i) and (ii) indicates two types of HPV identified in the same sample.

**Table 2 diagnostics-13-02765-t002:** Percentage frequency of HPV types identified among the HPV-positive married women in South Andaman.

Genus-Species	Genotype	Risk Group	Frequency (*n*)	Percentage (%)
Alpha-PV-9	HPV 16	High-Risk	32	57.2
Alpha-PV-9	HPV 52	High-Risk	1	1.8
Alpha-PV-9	HPV 58	High-Risk	2	3.6
Alpha-PV-6	HPV 66	High-Risk	3	5.3
Alpha-PV-9	HPV 33	High-Risk	1	1.8
Alpha-PV-7	HPV 18	High-Risk	3	5.3
Alpha-PV-11	HPV 73	High-Risk	2	3.5
Alpha-PV-6	HPV 53	High-Risk	1	1.8
Alpha-PV-6	HPV 30	Low-Risk	1	1.8
Alpha-PV-10	HPV 6	Low-Risk	2	3.6
Alpha-PV-3	HPV 61	Low-Risk	1	1.8
Alpha-PV-14	HPV 71	Low-Risk	1	1.8
Alpha-PV-3	HPV 81	Low-Risk	1	1.8
Alpha-PV-3	HPV 84	Low-Risk	2	3.6
Alpha-PV-3	HPV 87	Low-Risk	1	1.8
Un-typed	-	-	2	3.6

## Data Availability

The datasets of the current study are available from the corresponding author upon reasonable request.

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
