# Peer review of "Distribution of Human Papillomavirus Genotypes among the Women of South Andaman Island, India"

_diagnostics, 2023, doi:10.3390/diagnostics13172765_

Round 1

Reviewer 1 Report

This study conducted in Indian women who had age between 18-59 years old. What is the mean age? The HPV testing do not recommend to perform in women who had age less than 30 or 25 years depending on type of HPV testing.

This study was only prevalence study of sub-genotype of high risk HPV. HPV type 16 had two origin namely New York and Britain as shown in Table 1.

In part of keyword, the authors stated too much about cervical cancer that did not related in this study. The keyword should be only HPV, Andaman and Nicobar.

There was no demographic character ie mean age, coitarche, menarche, number sex partner and risk factors for cervical cancer. In discussion part, the author discussed about cervical cancer that did not involve in this study. This study was only prevalence of HPV in Andaman and Nicobar island only. The authors stated that this was the first study of HPV prevalence in Andaman and Nicobar island. This was inappropriate. There were many places in the world that did not study the subtype of HPV.

Reference

               Inhomogeneous pattern of reference ie reference # 12 and 13.

               Some references were not completed or presented in either scopus or pubmed search

Ref 30: not complete

Salavatiha Z, Shoja Z, Heydari N, Marashi SM, Younesi S, Nozarian Z, Jalilvand S. Lineage analysis of human papillomavirus type 18 based on E6 region in cervical samples of Iranian women. J Med Virol 2020; 92: 3815–20.

Author Response

Point-wise response to the Reviewer (R1):

Changes in the revised manuscript highlighted in yellow color.

Comments and suggestion

Response

1. This study conducted in Indian women who had age between 18-59 years old. What is the mean age? The HPV testing do not recommend to perform in women who had age less than 30 or 25 years depending on type of HPV testing.

The mean age ± SD of 1000 participants screened was 37.6 ±  9.2 years

1.UNICEF states that at least 1.5 million girls fewer than 18 get married in India. Nearly 16 per cent adolescent girls aged 15-19 are currently married. In addition, prohibition of child marriage act (PCMA), 2006 states that girls in India can get married at the age of 18, which is a legal age. (https://pib.gov.in/).

2. According to WHO, Women of reproductive age are 15-49 years.

Therefore, this study includes married women from 18 to 59 years of age.

Moreover this was a population based surveillance study done in this remote island to generate the information on the prevalence of HPV .So to have a more clearer picture, we have included women of 18-59 age group.

2. This study was only prevalence study of sub-genotype of high risk HPV. HPV type 16 had two origin namely New York and Britain as shown in Table 1.

Six sequences of HPV-16 found in South Andaman were closely related withHPV isolate found in New York and one HPV-16 isolate were related with HPV found in Britain as shown Table1.

3.In part of keyword, the authors stated too much about cervical cancer that did not related in this study. The keyword should be only HPV, Andaman and Nicobar.

Changes have been made in the revised manuscript

4.There was no demographic character ie mean age, coitarche, menarche, number sex partner and risk factors for cervical cancer. In discussion part, the author discussed about cervical cancer that did not involve in this study. This study was only prevalence of HPV in Andaman and Nicobar island only. The authors stated that this was the first study of HPV prevalence in Andaman and Nicobar island. This was inappropriate. There were many places in the world that did not study the subtype of HPV.

This paper mainly addresses the molecular characterization of HPV identified from this islands. The epidemiological data is under the process of publication, hence this data is not mentioned in the current manuscript.

This is a first comprehensive community based study done in the present geographical area (Andaman Island). Hence the statement is justified.

5. Inhomogeneous pattern of reference ie reference # 12 and 13.

Changes have been made in the revised manuscript

6. . Some references were not completed or presented in either scopus or pubmedsearch .

Ref 30: not complete

Salavatiha Z, Shoja Z, Heydari N, Marashi SM, Younesi S, Nozarian Z, Jalilvand S. Lineage analysis of human papillomavirus type 18 based on E6 region in cervical samples of Iranian women. J Med Virol 2020; 92: 3815–20.

Changes have been made in the revised manuscript

Reviewer 2 Report

  • The selected research area is innovative to some extent, but the sample size of the study is slightly small, and the gene polymorphism data is only elaborated without in-depth analysis.

The quality of English should been improved.

Author Response

Comments and Suggestions for Authors

Response

1.The selected research area is innovative to some extent, but the sample size of the study is slightly small, and the gene polymorphism data is only elaborated without in depth analysis

It could not see any major polymorphism in the sequence. L1 region as well as E6 region was short. To avoid the bias generating, we are performing the whole genomic sequencing. It is in progress and submitted as new manuscript.

Reviewer 3 Report

I read with attention the manuscript written by Parvez et al. about the distribution of human papillomavirus genotypes among a specific population: married women of South Andaman Island (India).

This topic is rarely covered in the literature since epidemiology of HPV in rural and isolated areas is poorly studied. This cross-sectional study is based on a significant population and the results are thus meaningful.

However, I found several structural and methodological flaws that must be revised for proper publication:

1.            Many abbreviations must be explicated at first use (GLOBOCAN, NFHS, …). Since the abbreviations are introduced, they must be used for the entire remaining text (HPV, HR…)

2.            Numbers are heterogeneously written as letters or figures. Please homogenize.

3.            Information contained in the paragraph from line 55 to line 68 must be related to relevant references.

4.            Methodology part should be divided in subparagraphs (population, inclusion, PCR, statistical analysis, …)

5.            Since the HPV exposure is dependent on sexual activity and multiple partners, the choice of only studying married women must be justified and discussed. This implies to justify why the woman mentioned in lines 94-96 were excluded

6.            The sampling method and pre-analytical process could skew the results, and this must be discussed. In fact, the samples were carried into PBS in cold temperature. This method does not protect the cells’ integrity and cell lysis occurred. Once in the lab, the samples were vortexed and centrifugated and the pellet was resuspended. If infected cells, known to be more fragile than normal cells, were lysed during transportation/conservation, the analysis could underestimate the prevalence because it focused on more normal cells and less infected cells. In this case, detection of housekeeping gene is not sufficient. Obviously, it is not possible to fix the problem, but it must be discussed.

7.            Lines 165-172: the results described the HPV genotypes according to frequency but the table 1 is based on risk and numerical order. To facilitate the reading, the authors should a concordant description of the results. Moreover, in the text, it would be relevant to mention the % of frequency of each genotype among infected women and in overall population.

8.            I may suggest explaining how the co-infections could be observed by using Sanger sequencing.

9.            Lines 191-207: the data are redundant, please synthetize.

10.          Lines 208-2013: these are also interesting results and it should be presented as figure 3 (or at least as supplemental figure)

11.          Line 222: prolonged HR-HPV infection is related to cancer so the authors must discuss the single detections they reported

12.          The discussion part must be revised . The study results are not compared to the literature and the authors do not discuss about how to interpret the results. The conclusion (line 275-278) is hasty and the process of drawing conclusions from the results is not described. Even if the study is cross-sectional and descriptive, the interpretation of the results should be less general and must open the debate to more precise outlooks.

Here are some additional and non-exhaustive minor revisions that support the above-mentioned revisions:

13.          Line 51 : use directly “HPV”

14.          Line 51 : “is members” ?

15.          Line 51: papillomaviridae  Papillomaviridae

16.          Line 52: explain “sds”

17.          Line 55: are? categorised

18.          Line 60: cause’s ?

19.          Line 74: Eight  8

20.          Line 111: explained PHCs and CHCs

21.          Line 128: the references #8 and #9 are not properly inserted

22.          Line 136: precise the number of samples instead of “few”

23.          Line 140: add details about the editor of MEGA11 software

24.          Line 158: four (04)  4

25.          Table 2: please precise that the “percentage” is actually the “percentage among women positive for HPV”

26.          Line 188: High-risk  HR

27.          Line 201: it is mentioned that “all” the HPV16 were lineage-A whereas the previous paragraph is about 13/14 lineage-A HPV16.

28.          Legend of figure 2: please explain the orange geometrical symbols

Minor editing (of English) is required

Author Response

Comments and Suggestions for Authors

Response

1. Many abbreviations must be explicated at first use (GLOBOCAN, NFHS, …). Since the abbreviations are introduced, they must be used for the entire remaining text (HPV, HR…)

As suggested, the abbreviations were explicated at first.

2. Numbers are heterogeneously written as letters or figures. Please homogenize.

As suggested, numbers were written homogenously.

3. Information contained in the paragraph from line 55 to line 68 must be related to relevant references.

As suggested by the reviewer, relevant references were included in the revised manuscript.

4. Methodology part should be divided in subparagraphs (population, inclusion, PCR, statistical analysis, …)

As per the suggestion, methodology part has been divided and included in revised manuscript.

5.  Since the HPV exposure is dependent on sexual activity and multiple partners, the choice of only studying married women must be justified and discussed. This implies to justify why the woman mentioned in lines 94-96 were excluded

Though HPV is transmitted through direct contact, most common mode of infection for HR-HPV is through sexual transmission. This was proved in many of the previous studies, worldwide.

Since the reproductive age of women is 15-49, choice of study participants were married women. In the country like India, It is restricted and will be ambiguous to elicit the data of sexual activity for unmarried women. Therefore, the choice of study participant were only married women.

The women mentioned in lines 94-96 were excluded for the following reasons.

1. Hysterectomy performed women cannot develop cervical cancer because of the removal of cervix.         

2. Routine cervical screening tests can usually be delayed in pregnant women until after they have had their baby. Pregnancy can make the result of the test harder to interpret and could be inaccurate.

3. Cytology screening in postmenopausal women has its limitation due to atrophic changes in the cervix.

6. The sampling method and pre-analytical process could skew the results, and this must be discussed. In fact, the samples were carried into PBS in cold temperature. This method does not protect the cells’ integrity and cell lysis occurred. Once in the lab, the samples were vortexed and centrifugated and the pellet was resuspended. If infected cells, known to be more fragile than normal cells, were lysed during transportation/conservation, the analysis could underestimate the prevalence because it focused on more normal cells and less infected cells. In this case, detection of housekeeping gene is not sufficient. Obviously, it is not possible to fix the problem, but it must be discussed.

In the present study we have used double stage sampling method. The sampling units were the villages or the municipal wards which were chosen after stratifying them in to urban and rural strata and randomly selected them as per the sample size required. Then by random sampling, 50 participants were selected from each of the selected sampling units. Finally, 700 women from rural and 300 from urban areas. We understand that there is possibility of skewed result. However the design effect of 2 was taken to adjust the sample size.

We agreed Reviewers comment, based on the published protocols for the use of PBS has been considered as an alternative collecting medium for HPV detection in the low-resource areas [Dankai W et al, 2021]. HPV is DNA virus and phosphate buffer saline can maintain pH and osmotic balance, as well as provide cells with water and essential inorganic ions [https://en.wikipedia.org/wiki/Phosphate-buffered_saline]. Since, the time duration required for the transportation of specimens to the laboratory was short and pelleting was done immediately. Overall, we assure that there were no compromise in the sample collection and processing for HPV.

Reference: Dankai W, Khunamornpong S, Siriaunkgul S, Soongkhaw A, Aithin P, Lekawanvijit S. An Evaluation of Phosphate Buffer Saline as an Alternative Liquid-Based Medium for HPV DNA Detection. Asian Pac J Cancer Prev. 2021;22(11):3441-3445. Published 2021 Nov 1. doi:10.31557/APJCP.2021.22.11.3441

7. Lines 165-172: the results described the HPV genotypes according to frequency but the table 1 is based on risk and numerical order. To facilitate the reading, the authors should a concordant description of the results. Moreover, in the text, it would be relevant to mention the % of frequency of each genotype among infected women and in overall population.

As suggested, changes have been made in the revised manuscript.

8. I may suggest explaining how the co-infections could be observed by using Sanger sequencing.

The data of the Real-Time PCR, PCR and sequencing of consensus L1 gene region this could identify different genotypes of HPV. In addition, sequencing for E6 and E7 gene of HPV16 as well as E6 gene for HPV18 was also done. The repeated sequencing and analysis in comparison with the results of Real-Time PCR, PCR could ensure the co-infections.

9. Lines 191-207: the data are redundant, please synthetize.

As suggested, two paragraphs has been modified and incorporated in the revised manuscript.

10. Lines 208-2013: these are also interesting results and it should be presented as figure 3 (or at least as supplemental figure)

Figure 2B depicted the Phylogenetic analysis for the E6 partial gene displayed the distribution of different Lineages of HPV-18 in South Andaman Island. However, E7 gene sequences of HPV-18 were not available and we modified the result accordingly.

We have represented the Phylogenetic analysis for the E7 partial gene displayed the distribution of different Lineages of HPV-18 in South Andaman Island (Figure 3).

11. Line 222: prolonged HR-HPV infection is related to cancer so the authors must discuss the single detections they reported.

Prolonged or persistent HPV infection is related to cervical cancer and dyslasias,But due to the cross sectional nature of the study, persistent infection could not be assessed. As it need follow up study of the HPV infected cases for atleast 2 years.

However due to the non-relevance of line 222 of discussion to the findings in the manuscript, this line is removed.

12. The discussion part must be revised. The study results are not compared to the literature and the authors do not discuss about how to interpret the results. The conclusion (line 275-278) is hasty and the process of drawing conclusions from the results is not described. Even if the study is cross-sectional and descriptive, the interpretation of the results should be less general and must open the debate to more precise outlooks.

As suggested, discussion and conclusion part has been revised.

13. Line 51 : use directly “HPV”

Suggestion has been revised.

14. Line 51 : “is members” ?

Suggestion has been revised.

15. PapillomaviridaeàLine 51: papillomaviridae

Changes have been made in the revised the manuscript.

16. Line 52: explain “sds”

ds- Double stranded. Mistake has been changed and made in the revised manuscript.

17. Line 55: are? Categorized

Changes have been made in the revised manuscript.

18. Line 60: cause’s ?

Changes have been made in the revised the manuscript.

19.  8àLine 74: Eight

Suggestion has been revised.

20. Line 111: explained PHCs and CHCs

Primary health care (PHC) ensures people receive quality comprehensive care - ranging from promotion and prevention to treatment, rehabilitation and palliative care - as close as feasible to people’s everyday environment.

Community Health Centre (CHC) is located at block/division level and serves as a referral centre for PHCs. It is to be staffed by medical officers and Midwives.

21. Line 128: the references #8 and #9 are not properly inserted

As suggested, the references were properly inserted to the revised manuscript.

22. Line 136: precise the number of samples instead of “few”

Changes have been made in the revised manuscript.

23. Line 140: add details about the editor of MEGA11 software

The sequencing analysis was performed by,

Dr. Nagarajan Muruganandam, Scientist-D and corresponding author, ICMR-RMRC, Port Blair, Andaman and Nicobar Islands.

24. Line 158: four (04)

Changes have been made in the revised manuscript

25. Table 2: please precise that the “percentage” is actually the “percentage among women positive for HPV”

Changes have been made in the revised manuscript

26. HRàLine 188: High-risk

Changes have been made in the revised manuscript

27. Line 201: it is mentioned that “all” the HPV16 were lineage-A whereas the previous paragraph is about 13/14 lineage-A HPV16.

Changes have been made in the revised manuscript

28. Legend of figure 2: please explain the orage geometrical symbols

Changes have been made in the revised manuscript

Round 2

Reviewer 1 Report

Abstract part:

Manuscript: Global burden of cervical cancer associated with various HPV sub-lineages.

Comment: This was not the result of this study. It should be stated in the discussion part.

Manuscript: HPV-16 A1 sub-lineage was globally widespread whereas sub-lineages A1, A2 and D1 was prevailing in South Andaman.

Comment: It should be stated only “sub-lineages A1, A2 and D1 of HPV type 16 were prevailing in South Andaman “

Introduction part: There was still grammatic error in the manuscript. The native English speaker was needed to audit this manuscript. The introduction part was still too much. The study was only descriptive study for prevalence of HPV only.

Line 42: What is the reference of “In Asia, the incidence rate of cervical cancer was 58.2 percent.”

This statement was not corrected as suggestion in the previous review.

Result part

Manuscript: Out of 1000 samples screened, 50 specimens tested positive for HPV L1 gene amplification. Subsequently, type-specific PCR for HR-HPV16 1 and 18 identified 32 patients positive for HR-HPV16 and four patients positive for HR-HPV18. DNA sequencing for the L1 region was successful for 24 samples.

Comment: Prevalence of positive HPV was 5 (50/1000) percent. HPV 16 and 18 were 3.2 and 0.4 percent, respectively. What is the meaning of 24 samples from this paragraph.

Reference: There were still multiple patterns of reference.

Reference #2 and 3 were invalid.

Reviewer 3 Report

The authors revised the manuscript according to the comments from the reviewing. This satisfies me